# Entropy-Driven Mixed-Precision Quantization for Deep Network Design

**Zhenhong Sun**[1]     **Ce Ge**[1]     **Junyan Wang**[1,2,*]     **Ming Lin**[1,3*]

**Hesen Chen**[1]     **Hao Li**[1]     **Xiuyu Sun**[1†]

[1]Alibaba Group     [2]University of New South Wales     [3]Amazon.com, Inc

## Abstract

Deploying deep convolutional neural networks on Internet-of-Things (IoT) devices is challenging due to the limited computational resources, such as limited SRAM memory and Flash storage. Previous works re-design a small network for IoT devices, and then compress the network size by mixed-precision quantization. This two-stage procedure cannot optimize the architecture and the corresponding quantization jointly, leading to sub-optimal tiny deep models. In this work, we propose a one-stage solution that optimizes both jointly and automatically. The key idea of our approach is to cast the joint architecture design and quantization as an Entropy Maximization process. Particularly, our algorithm automatically designs a tiny deep model such that: 1) Its representation capacity measured by entropy is maximized under the given computational budget; 2) Each layer is assigned with a proper quantization precision; 3) The overall design loop can be done on CPU, and no GPU is required. More impressively, our method can directly search high-expressiveness architecture for IoT devices within less than half a CPU hour. Extensive experiments on three widely adopted benchmarks, ImageNet, VWW and WIDER FACE, demonstrate that our method can achieve the state-of-the-art performance in the tiny deep model regime. Code and pre-trained models are available at https://github.com/alibaba/lightweight-neural-architecture-search.

## 1   Introduction

With the attention of many intelligent IoT applications that have grown recently, the demand for low-cost and low-energy IoT devices is significantly increasing, such as tiny NPUs and microcontroller units (MCUs). These devices have tight-resources like hundreds KB on-chip memory (SRAM), MB-level storage (Flash) and low speed computing ability. Thus, how to design and deploy efficient tiny deep models for IoT devices is becoming a main topic in Tiny Machine Learning (TinyML) [1].

As most IoT devices have very limited on-chip memory, one of the most critical challenges in TinyML is to control the peak memory during inference. Some works re-design the network architecture to maintain the peak memory occupation [27, 39, 10, 17, 33, 16] by network transformed techniques, in which an accessible and efficient technique is to slice input images into multiple patches so that the inference can be split into multiple independent processes to meet the peak memory constraint for once calculation. In addition, Neural Architecture Search (NAS) is a popular tool to re-design tiny deep models automatically [10, 17, 33, 16] with an elaborate search space. After re-designing the

---

*Work done in Alibaba Group.

†Correspondence to: Xiuyu Sun <xiuyu.sxy@alibaba-inc.com>.

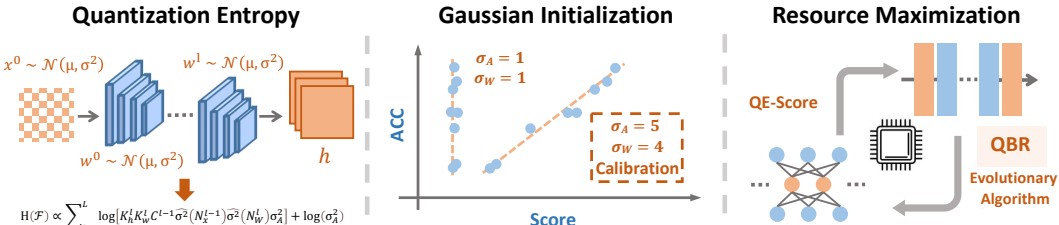

Figure 1: Illustration of our proposed strategy for deep network design on IoT devices, which consists of quantization entropy score, Gaussian initialization calibration, and resource maximization.

neural architecture, the promotable stage is to reduce the memory occupation via mixed-precision quantization [34, 36, 3, 11, 35, 10, 26, 38, 4]. Nevertheless, the incoherence of such a two-stage design procedure leads to the inadequate utilization of resources, therefore producing sub-optimal models within tight resource requirements for IoT devices.

To compensate for the sub-optimality of the two-stage design, this work studys how to optimize the network architecture jointly with mixed-precision quantization in a more efficient manner. Recently, training-free based approaches [22, 6, 32, 18, 31] have emerged for neural architecture search, accelerating the progress of the model design. Owing to constructing an alternative proxy to rank candidate networks instead of a training-based accuracy indicator, they achieve commendable effectiveness for the full-precision model design. Hence, considering the progress of the proxy mechanism, training-free technology could provide hands-down feasibility for integrated architecture searching, but there is still a lack of key techniques for cooperating mixed-precision quantization. Thus, we explore advancing the training-free technology to optimize both architecture and quantization in a one-stage lightweight way.

Inspired from [31], a deep neural network can be regarded as an information system, and the information of the system can be proportional to the entropy of the last feature map, which reflects the expressiveness of the network. Further, we reformulate the informationalized network with mixed-precision quantization and measure its entropy as the proxy for the performance of the quantization network. Concretely, we first comprehend the quantization effect on the entropy of the information system and propose a Quantization Entropy Score (QE-Score) to calculate the entropy. Then, according to the study of how Gaussian initialization on variables affects the value of QE-Score, we design a grid calibration process to determine the Gaussian initial values. Finally, a Quantization Bits Refinement (QBR) strategy embedded in the evolutionary algorithm is introduced to maintain higher-precision weights and lower-precision activations in the front few-layer, while the opposite is in the latter few layers. The entire process is illustrated in Figure 1.

The key contributions of our work are summarized as follows:

- To the best of our knowledge, we first present the ranking strategy of mixed-precision quantization networks in the entropy view. And Quantization Entropy Score is proposed with a calibrated initialization to measure the expressiveness of the network.

- Quantization Bits Refinement is proposed to adjust mixed quantization bits, which can ensure that each layer is assigned with proper quantization precision, and maximize the utilization of memory and storage resources on the IoT devices.

- Benefitting from the QE-Score, our approach can achieve architecture searching within less than half a 64-core CPU hour. Extensive experiments demonstrate that our searched model can achieve state-of-the-art performance on IoT devices for both classification and detection tasks.

## 2   Related Work

**Tiny Deep Models.** Recent attempts [27, 39, 10, 17, 33, 16] have been made to design efficient neural networks by reducing the peak memory of the front layers intuitively. Some works [27, 39, 16] aim to reduce the resolution of the front layers by slicing the image to several patches. For instance, RaScaNet [39] learns the representation of the whole image using a recurrent neural network through a raster-scanning image reading pipeline. Others works [10, 17, 33, 16] focus on investigating Neural

Architecture Search (NAS) to design architecture under the peak memory budget of IoT devices. For example, MCUNet series [17, 16] address that automatically optimizes the search space to adapt the tiny resource constraints. Then, NAS search is performed for handling the tiny and diverse memory constraints on various microcontrollers. These works re-design the network architecture to maintain the peak memory occupation, however the mixed-precision quantization strategy is not considered to further reduce resource utilization.

**Mixed-precision Quantization.** Mixed-precision quantization search [34, 36, 3, 11, 35, 10, 26, 38, 4] can reduce the peak memory, flash storage and computation cost of neural networks. Some methods [34, 36, 26, 4] focus on mixed-precision quantization on specific models (i.e., ResNet [13] and MobileNetV2 [28]) without jointly combining with the architecture design. For example, DNAS [36] formulates quantizing different layers with different bit-widths as a neural architecture search problem, and proposes a novel differentiable neural architecture search framework to efficiently explore it with gradient-based optimization. Other works [3, 11, 35, 10, 38] attempt to jointly optimize the network architecture on cloud and mobile models and quantization by training each candidate network or a big supernet on various candidate architectures. For instance, APQ [35] conducts a joint search between neural architecture design, pruning policy, and quantization policy, and proposes a predictor-transfer method to tackle the high cost of quantization-aware accuracy training. However, the architecture design in these methods pays less attention to the constraints of the IoT devices. Therefore, how to jointly design the tiny deep models and do mixed-precision quantization is still worth exploring.

**Training-free NAS methods.** Some attempts [22, 6, 32, 18, 31] proposed training-free strategies for architecture searching, which construct an alternative proxy to rank the initialized networks without training. For example, SynFlow [32] preserves the total flow of synaptic strengths through the network at initialization, subject to a sparsity constraint as the proxy. Specially, Zen-NAS [18] uses the gradient norm of the input image as ranking score, and MAE-DET [31] estimates the differential entropy of the last feature map to represent the expressiveness of a network, based on the maximum entropy theory [24]. However, these methods construct the proxy on full-precision models, and can't directly indicate the correlation between mixed-precision quantization and accuracy. To address the above issues, our work aims to optimize the network architecture jointly with mixed-precision quantization in one-stage, considering the lightweight neural architecture search approach.

## 3 Problem Statement

### 3.1 Constraints of IoT Devices

Although significant research has been done to investigate architecture searching for IoT devices, designing models under limited resources remains a challenging issue. Limited on-chip SRAM memory and Flash storage are the major constraints [12, 17, 16] for deploying deep learning models on IoT devices, especially for MCUs. For example, a state-of-the-art ARM Cortex-M7 MCU merely has 512kB SRAM and 2MB Flash, which is impossible to run the off-the-shelf ResNet-50 [13] or MobileNetV2 [28]. As a typical tiny model, full-precision MobileNetV2 needs 5.6M peak memory and 13.5M storage, while the int8-precision version needs 1.4M peak memory and 3.4M storage [17], which far exceeds the limitation of mainstream IoT devices. The distribution of memory and model size of MobileNetV2 is illustrated as shown in Figure 2. On the other hand, the limited resource budget might be satisfied by lower precision quantization, but the performance of the models will drop more.

### 3.2 Low-precision Quantization

To analyze the performance drop, we conduct an overall low-precision quantization [9] on MobileNetV2 except for the first and last layers, which follow the common practice of using 8-bit for high accuracy [25, 42, 2, 9]. As shown in Table 1 and Figure 3, we observe that **lower bit precision lower accuracy**. Only "A3W2" model fits both 512KB memory limit and 2MB storage limit with 23.17% ACC decline, which uses 3-bit for all activations and 2-bit for all weights. However, in "A3W2" Model, the activations in latter stages and the weights in front stages can utilize larger bits quantization, which change less memory and storage according to the imbalanced resource utilization in Figure 3. Thus, the resource utilization of fixed-precision quantization is not high in some stages, which means we need mixed-precision quantization to improve accuracy. Because mixed-precision

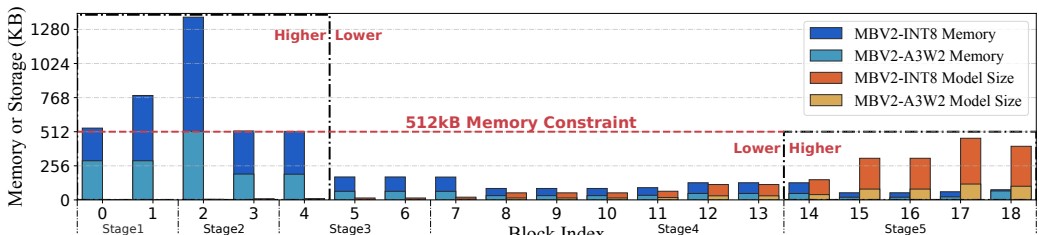

Figure 2: Imbalanced memory and model size distribution on MobileNetV2 [28]. Top five blocks determine the memory bottleneck of the entire network due to high resolution, and last five blocks occupy 80% of model size due to higher output channels. "A3W2" model could satisfy both 512KB memory limit and 2MB storage limit, which uses 3-bit for activations and 2-bit for weights.

quantization uses lower bit on tight-resource position and higher bit on rich-resource position to ensure precision stability. Therefore, we should apply neural architecture search on mixed-precision quantization for selected IoT devices.

Table 1: TOP-1 ACC of fixed-precision MobileNetV2 models on ImageNet with 120 training epochs. Bold values meet the 512KB SRAM limit, and underline values meet the 2MB Flash limit.

| Activation | Weight Bit | | | | | |
|---|---|---|---|---|---|---|
| Bit | 2 | 3 | 4 | 5 | 6 | 8 |
| 3 | **_47.43_** | **59.38** | **62.78** | **63.59** | **64.06** | **64.28** |
| 4 | _55.54_ | 64.74 | 67.78 | 68.50 | 68.59 | 68.94 |
| 5 | _56.66_ | 66.31 | 69.23 | 69.75 | 69.99 | 70.25 |
| 6 | _57.73_ | 66.62 | 69.11 | 70.00 | 70.07 | 70.48 |
| 8 | _57.89_ | 66.69 | 69.25 | 70.02 | 70.17 | 70.60 |

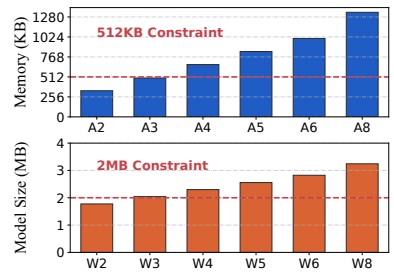

Figure 3: Peak memory and model size of fixed-precision MobileNetV2 models.

# 4 Quantization Entropy

In this section, we propose a Quantization Entropy Score for measuring the expressiveness of the mixed-precision quantization model.

## 4.1 Maximum Entropy for Full-precision Models

According to the Maximum Entropy Principle [15] and its successful deep learning applications [29, 5, 40, 31], a deep neural network can be regarded as an information system, and the differential entropy of the last output feature map represents the expressiveness of the system. Given by the following theorem:

**Theorem 1.** *For any continuous distribution $\mathbb{P}(x)$ of mean $\mu$ and variance $\sigma^2$, its differential entropy is maximized when $\mathbb{P}(x)$ is a Gaussian distribution $\mathcal{N}(\mu, \sigma^2)$.*

The differential entropy of a distribution is upper bounded by a Gaussian distribution with the same mean and variance. Suppose $x$ is sampled from Gaussian distribution $\mathcal{N}(\mu, \sigma^2)$, the differential entropy [24] $H$ of $x$ is then:

$$H(x) = \int_{-\infty}^{+\infty} -\log(p(x))p(x)\,dx \quad \propto \log(\sigma^2), \tag{1}$$

where $p(x)$ represents the probability density function of $x$. Moreover, the entropy $H$ represents the expressiveness of a deep system, which correlated with the performance of a deep neural network [31]. Specifically, given a convolutional network with $L$ layers of weights $\mathbf{W}^1, ..., \mathbf{W}^L$, which are initialized from a standard Gaussian distribution, the forward inference is given by:

$$\boldsymbol{x}^l = \phi(\mathbf{W}^l * \boldsymbol{x}^{l-1}) \quad \text{for } l = 1, \dots, L, \tag{2}$$

where $x^l$ denotes the $l^{th}$ layer feature map and $\phi(\cdot)$ represents the activation function. For simplicity, the bias of the convolutional layer is set to zero and the activation function is omitted. As we initialize the input $x^0$ from Gaussian distribution, the upper bound entropy $H(\mathcal{F})$ of the network $\mathcal{F}$ is proportional to $\log(\sigma^2(x^L))$, where $\sigma^2(x^L)$ represents the variance of the $x^L$ and is computed by the forward inference. To this end, we can measure the expressiveness of full-precision networks without training and inference. The detailed description can be found in [31].

Further, according to the *product law of expectation* [23] and *Bienaymé's identity in probability theory* [20], we can obtain the expectation and variance of element $i$ of $x^1$ layer feature map in our system by:

$$\mathbb{E}(\boldsymbol{x}_i^1) = 0, \quad \sigma^2(\boldsymbol{x}_i^1) = \sum_{h=1}^{K_h^1} \sum_{w=1}^{K_w^1} \sum_{c=1}^{C^0} \left[ \sigma^2(\boldsymbol{x}_{chw}^0) \times \sigma^2(\boldsymbol{W}_{chw}^1) \right], \tag{3}$$

where $\{K_h^1, K_w^1\}$ represents the kernel size of the first layer in the CNN, and $C^0$ denotes its input channels size. Further extend to the $l$-layer convolution, we can obtain corresponding layer's expectation and variance:

$$\mathbb{E}(\boldsymbol{x}_i^l) = 0, \quad \sigma^2(\boldsymbol{x}_i^l) = \sum_{h=1}^{K_h^l} \sum_{w=1}^{K_w^l} \sum_{c=1}^{C^{l-1}} \left[ \sigma^2(\boldsymbol{x}_{chw}^{l-1}) \times \sigma^2(\boldsymbol{W}_{chw}^l) \right]. \tag{4}$$

The detailed derivation process is present in **Appendix A**. Next, we will explore how to introduce mixed-precision quantization into the calculation process.

### 4.2 Quantization Entropy for Mixed-Precision Models

Mixed-precision quantization to the network $F$ requires inserting low-precision conversion behind Gaussian initialized input and weights. Learning from the practices of entropy coding [30, 24] with fixed quantization step size of 1, given the variable $R$ to quantize symmetrically, the conversion of $N$-bit quantization representation [26, 9] can be defined as:

$$Q = \text{round}(\text{clamp}(R, -2^{N-1}, 2^{N-1} - 1)), \tag{5}$$

where clamp returns $R$ with values below $-2^{N-1}$ set to $-2^{N-1}$ and values above $2^{N-1} - 1$ set to $2^{N-1} - 1$. According to the work of [31], the input of each layer is zero-mean distribution when deriving the entropy, so that the upper bound of $Q$ is set as $2^{N-1}$. Then, the variance of the quantization value $Q$ can be given by:

$$\hat{\sigma}^2(N) = \sum \left[ \mathbb{P}(Q) \times (Q - \mu_Q)^2 \right], \quad \mathbb{P}(Q) = \int_{R_l}^{R_r} \mathbb{P}(R), \tag{6}$$

where $Q \in [-2^{N-1}, 2^{N-1}]$ and $\hat{\sigma}^2$ denotes the truncated variance of $Q$, $\mathbb{P}(Q)$ is the probability of $Q$ relied on the distribution of $R$. Therefore, $\mathbb{P}(Q)$ equals to: (1) $\int_{-\infty}^{-2^{N-1}+0.5} \mathbb{P}(R)$, when $Q = -2^{N-1}$; (2) $\int_{Q-0.5}^{Q+0.5} \mathbb{P}(R)$, when $-2^{N-1} < Q < 2^{N-1}$; (3) $\int_{2^{N-1}-0.5}^{\infty} \mathbb{P}(R)$, when $Q = 2^{N-1}$.

As shown in Figure 4, we can figure out that the quantization of Gaussian variable will decrease the variance of the variable. Given a Gaussian distribution with $\sigma$ and $N$ bit precision, the decrease is fixed, so that we can pre-compute the decrease, which will speed up the computation of Eq. 6. The pre-computed quantization standard deviation $\hat{\sigma}(N)$ is demonstrated in Table 2.

Then, apply the quantization to Eq. 4, the $\sigma^2(x_j)$ of $j^{th}$ layer become larger during the depth increase. As we set the quantization step as 1 in Eq. 5, the distribution in Figure 4 will be much smoother, and the probability will close to 0. So that the different bits of quantization can not be distinguished in the next layer. If we use a dynamic quantization step in each layer to avoid this phenomenon, the computation will be more complex. Therefore, we adopt a scaling parameter $\sigma_S^2(x_{chw}^{l-1})$ to normalize the $x_j$ to $\sigma_A$ instead of adjusting the quantization step. Following the weights obey $\mathcal{N}(0, \sigma_W)$ [18, 31], then use $N_x^{l-1}$ and $N_W^l$ bit to quantize the $l$-layer's input and weight respectively, Eq. 4 can be revised as

$$\sigma^2(\boldsymbol{x}_i^l) = \sum_{h=1}^{K_h^l} \sum_{w=1}^{K_w^l} \sum_{c=1}^{C^{l-1}} \left[ \hat{\sigma}^2(N_x^{l-1}) \times \hat{\sigma}^2(N_W^l) \right] \times \sigma_S^2(\boldsymbol{x}_{chw}^{l-1}), \quad \sigma_S^2(\boldsymbol{x}_{chw}^{l-1}) = \sigma^2(\boldsymbol{x}_{chw}^{l-1})/\sigma_A^2. \tag{7}$$

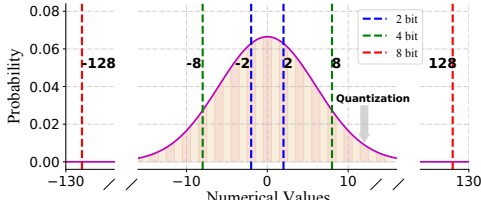

Figure 4: $N = \{2, 4, 8\}$ bit quantization on Gaussian variable. The upper and lower bounds represent truncation. Shaded areas represent quantization.

Table 2: Look up table of $\hat{\sigma}(N)$ according to $\sigma$ and $N$ bit. Low-precision leads to small variance.

| $\sigma$ | $N$ Bit Precision | | | | | | |
|---|---|---|---|---|---|---|---|
| | 2 | 3 | 4 | 5 | 6 | 7 | 8 |
| 1 | 1.00 | 1.04 | 1.04 | 1.04 | 1.04 | 1.04 | 1.04 |
| 2 | 1.47 | 1.94 | 2.02 | 2.02 | 2.02 | 2.02 | 2.02 |
| 4 | 1.73 | 2.89 | 3.85 | 4.01 | 4.01 | 4.01 | 4.01 |
| 6 | 1.82 | 3.26 | 5.04 | 5.96 | 6.00 | 6.00 | 6.00 |

Subsequently, the variance of $\boldsymbol{x}^L$ can be obtained by propagating the variances from previous layers:

$$\sigma^2(\boldsymbol{x}_i^L) = \sigma^2(\boldsymbol{x}_{chw}^0) \times \prod_{l=1}^{L} K_h^l K_w^l C^{l-1} \hat{\sigma}^2(N_x^{l-1}) \hat{\sigma}^2(N_W^l)/\sigma_A^2, \tag{8}$$

where $\boldsymbol{x}^0$ is initialized by $\sigma_A^2$. Refer to the Eq. 1, the upper bound entropy $H(\mathcal{F})$ is proportional to:

$$H(\mathcal{F}) \propto \sum_{l=1}^{L} \log \left[ K_h^l K_w^l C^{l-1} \hat{\sigma}^2(N_x^{l-1}) \hat{\sigma}^2(N_W^l)/\sigma_A^2 \right] + \log(\sigma_A^2), \tag{9}$$

where $\hat{\sigma}^2(N_x^l)$ and $\hat{\sigma}^2(N_W^l)$ are calculated by Eq. 6 with initial standard deviation $\sigma_A$ and $\sigma_W$. We name the value of Eq. 9 as **Quantization Entropy Score** (QE-Score), and detailed derivation is introduced in Appendix A. As the QE-Score only depends on the structural parameters of the network, quantization function, and initial standard deviation $\sigma_A$ and $\sigma_W$, we can compute QE-Score on CPUs to rank candidate networks during search, which greatly reduces the dependence on hardware. In the next section, we will show how to determine the initial standard deviation $\sigma_A$ and $\sigma_W$.

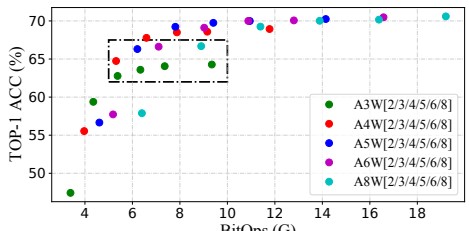

Figure 5: ACC vs. BitOps. "AXWY" represents X-bit for all activations and Y-bit for all weights. A4W3 model has better accuracy than A3W4 model.

### 4.3 Gaussian Initialization Calibration

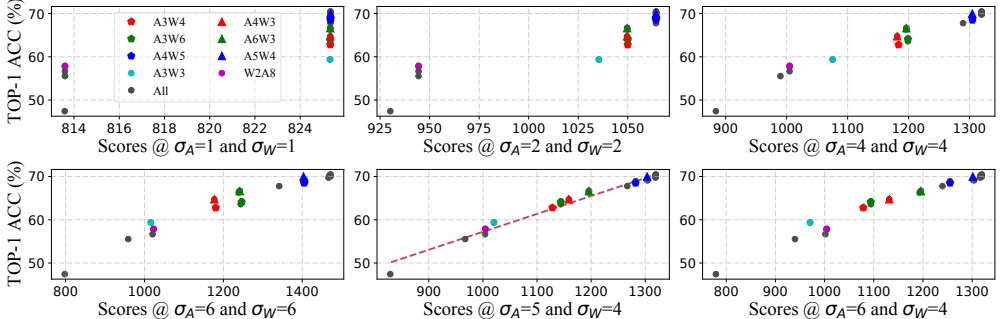

Figure 6: Gaussian initialization calibration between the accuracy and QE-Score on fixed-precision MobileNetV2 models.

We apply the commonly used MobileNetV2 architecture with fixed-precision quantization for calibration, and these models have been summarized in Table 1. According to the relationship between accuracy and computation in Figure 5 and the pre-computed quantization standard deviation $\hat{\sigma}(N)$ in

Table 2, we can figure out that: 1) Increasing the initial standard deviation $\sigma$ will help us distinguish the disparity of entropy between different bits; 2) Quantization on activations and weights has different effects on accuracy. Therefore, we set different values of $\sigma_A$ and $\sigma_W$ to show differences between activations and weights. In summary, we use a grid calibration process to seek for the appropriate initial $\sigma_A$ and $\sigma_W$, and demonstrate the calibration process in Figure 6.

According to Figure 6, we observe that when gradually increasing the values of $\sigma_A$ and $\sigma_W$, QE-Scores are gradually positive correlated with accuracy until $\sigma_A = \sigma_W = 4$. Then, we adjust $\sigma_A = 5$ and $\sigma_W = 4$ to rank the diversity of activations and weights on accuracy. Benefit from the Gaussian initialization calibration, QE-Score can help us rank various architectures without training and evaluating during search, and design a high-expressiveness architecture for IoT devices.

## 4.4 Resource Maximization for IoT Devices

We apply Evolutionary Algorithm (EA) along with low-precision quantization to obtain the optimal architecture by the proposed QE-Score for IoT devices. Based on the observation in Figure 2, the front few layers dominate the peak memory of the entire model due to high resolution, and the distribution of model weights generally shows an upward trend with the increase of feature map channels. Considering the tight SRAM memory and Flash storage constraints of IoT devices, maintaining higher-precision weights and lower-precision activations in the front few layers is an eligible strategy, while it is the opposite in the latter few layers. Therefore, we propose a quantization bits refinement strategy (Figure 7) embedded in the EA to realize it, to assess the bit distribution and make appropriate adjustments to maximize resources utilization. **The detailed algorithm is introduced in Appendix B.**

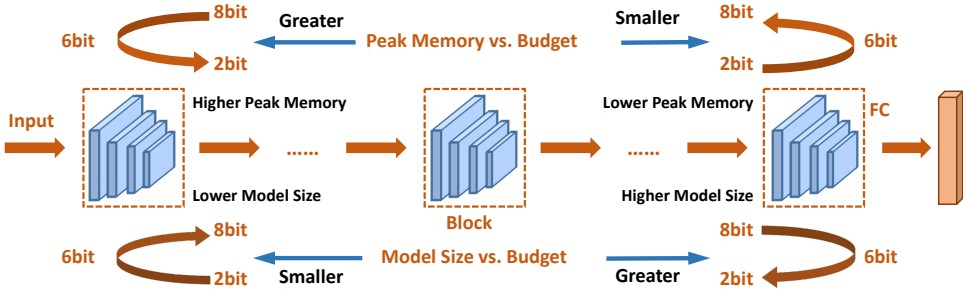

Figure 7: Quantization Bits Refinement (QBR) strategy for a candidate architecture to redistribute the mixed-precisions. For activations, we scale the mixed-precision to make the peak memory meet the budget. Accordingly, we also increase or decrease the mixed-precision of the weights from the smallest blocks or the largest block separately to guarantee the model size approaches the budget. Orange curve arrows in the four corners mean the adjustment scale of the precision value from 2 bit to 8 bit or 8 bit to 2 bit.

## 5 Experiments

### 5.1 Implementation Details

**Searching Settings**. The evolutionary algorithm and super-block definition follow the work of [18, 31]. The evolutionary population $N$ is set as 512 with total 500000 iterations. To compare with common mix-quantization search [34, 36, 11, 35] and IoT device applications [17, 16, 27, 39], we build our network with MobileNetV2-based blocks, using the inverted bottleneck block with expansion ratio in $[1.5, 6]$. Specifically, low-precision values are random selected from $\{2, 3, 4, 5, 6, 8\}$ in our search, and three layers in each block share the same precision value.

**Datasets and Training Settings**. We use two standard benchmarks in this work: ImageNet [8] and Visual Wake Words (VWW) [7]. For ImageNet-1K dataset, our models are trained for 240 epochs without special indication. All models are optimized by SGD with a batch size of 512 and Nesterov momentum factor of 0.9. Initial learning rate is set to 0.4 with cosine learning rate scheduling [21], and the weight decay is set to 4e-6. Besides, to validate the expansibility of our method, we further

evaluate it on the WIDER FACE [37] object detection dataset. When training with low-precision values, we follow the implementation in [9] to do the quantization. For VWW and WIDER FACE, we follow the training settings in [16]. **More implementation details are introduced in Appendix C.**

## 5.2 Mixed-Precision Comparison

To compare with other mixed-precision search methods [34, 36, 11, 35], we conduct our searching on the MobileNetV2-level under the computation budget (BitsOps), and the BitsOps budget is 19.2G to compare with MobileNetV2-8bit, and 7.0G for MobileNetV2-4bit.

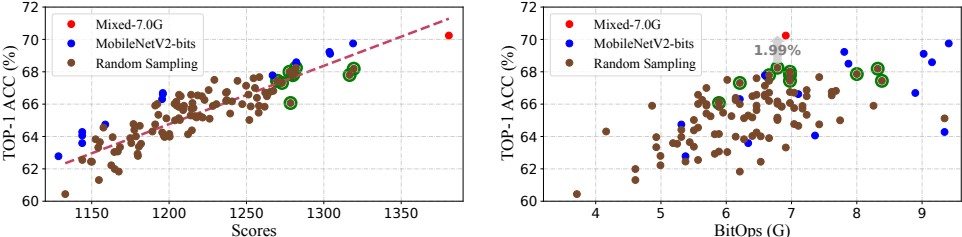

Figure 8: Correlations between TOP-1 accuracy and QE-Score with 100 random sampled networks. Green circle points represent TOP-10 score models in all random samples.

**Random Correlation Study.** We randomly selected 100 models without QE-Score under the same searching space to verify the correlation between our QE-Score and accuracy on ImageNet, before comparing with SOTA methods. Results are shown in Figure 8, and we can observe that: 1) The proposed QE-Score is closer to the linear growth relationship with TOP-1 accuracy than BitOps; 2) The searched mixed-7.0G model outperforms the best random model with TOP-1 accuracy of nearly 2%. According to these observations, our QE-Score is valid to rank various architectures instead of training and evaluating.

**Comparison with SOTA Models.** Table 3 presents the results of fixed- and mixed- precision searching. Obviously, our models outperform state-of-the-art mixed-precision methods. In particular, our mixed-19.2G model has 2.9% accuracy boost than MobileNetV2-8bit baseline, and our mixed-7.0G model has better accuracy than MobileNetV2-4bit baseline (from 68.9% to 70.8%). Note that, as we apply formula calculation rather than inference, our QE-Score can use CPU instead of GPU. This will significantly reduce the dependence on hardware. More impressively, the marginal cost of $CO_2$ emission of our method is two orders of magnitudes smaller than other work, benefiting from computational efficiency.

Table 3: Comparison with state-of-the-art efficient models with mixed-precision quantization. MBV2-4bit use 4-bit for the overall layers except for the first and last layer. †: 64 cores of Intel(R) Xeon(R) Platinum 8269CY CPU @ 2.50GHz.

| Model | Quant. | Search Devices | Design Cost (hours) | Model Size (MB) | BitOps (G) | ImageNet TOP-1 | CO2e (marginal) |
|---|---|---|---|---|---|---|---|
| MBV2 [28] | 8-bit | - | - | 3.4 | 19.2 | 71.9% | - |
| MBV2 [28] | 4-bit | - | - | 2.3 | 7.0 | 68.9% | - |
| MBV2+HAQ [34] | mixed | GPUs | 96N | - | - | 71.9% | 27.23N |
| DNAS [36] | mixed | GPUs | 300N | - | 57.3 | 74.0% | 11.34N |
| SPOS [11] | mixed | GPUs | 288+24N | - | 51.9 | 74.6% | 82+6.81N |
| APQ [35] | mixed | GPUs | 2400+0.5N | - | 16.5 | 74.1% | 672+0.14N |
| APQ [35] | mixed | GPUs | 2400+0.5N | - | 23.6 | 75.1% | 672+0.14N |
| Ours-19.2G | mixed | CPUs† | 0.5N | 3.2 | 18.8 | **74.8%** | 0.19N |
| Ours-7.0G | mixed | CPUs† | 0.5N | 2.2 | 6.9 | **70.8%** | 0.19N |

## 5.3 Tiny Image Classification

**Large-scale Classification on ImageNet.** We searched tiny models under three hardware specifications, and comparative results with recent state-of-the-art tinyML solutions are shown in Table 4. We can figure out that our model outperforms others on all three SRAM/Flash budgets. Under the tight constraints of 256kB SRAM and 1MB Flash, our model significantly improves the TOP-1 accuracy by 6.2% and 4.5% over 8-bit and 4-bit quantized MCUNet, respectively. Under 512kB SRAM and 1MB Flash, our model achieves a new record of 72.8% TOP-1 accuracy. We think these remarkable achievements benefit from improved SRAM/Flash utilization, which means more accurate mixed-precision quantization can specialize higher-capacity structures on resource-constrained IoT devices. Note that MCUNetV2 employed patch-based inference scheduling to reduce memory usage, which is an orthogonal technique to mixed-precision quantization.

Table 4: Comparison of ImageNet classification accuracy on IoT devices

| Model | Quant. | 256kB SRAM, 1MB Flash | | | 320kB SRAM, 1MB Flash | | | 512kB SRAM, 2MB Flash | | |
|---|---|---|---|---|---|---|---|---|---|---|
| | | Mem | Size | Acc. | Mem | Size | Acc. | Mem | Size | Acc. |
| MBV1 [14, 26] | mixed | <256kB | <1MB | 60.2% | - | - | - | <512kB | <2MB | 68.0% |
| MBV2 [28] | 8-bit | - | - | - | 308kB | 0.72MB | 49.0% | - | - | - |
| Proxyless [3] | 8-bit | - | - | - | 292kB | 0.72MB | 56.2% | - | - | - |
| MCUNet-int8 [17] | 8-bit | 238kB | 0.70MB | 60.3% | 293kB | 0.70MB | 61.8% | 452kB | 1.65MB | 68.5% |
| MCUNet-int4 [17] | 4-bit | 233kB | 0.67MB | 62.0% | 282kB | 0.67MB | 63.5% | 498kB | 1.56MB | 70.7% |
| MCUNetV2 [16] | 8-bit | 196kB | 0.79MB | 64.9% | - | - | - | 465kB | 1.67MB | 71.8% |
| Ours | mixed | 253kB | 0.73MB | **66.5%** | 308kB | 0.71MB | **68.2%** | 507kB | 1.67MB | **72.8%** |

**Low-energy Application on Visual Wake Words.** The Visual Wake Words dataset (VWW) can represent a realistic IoT use-case of identifying the presence of persons. Experimental results on VWW dataset are shown in Figure 9. We can observe that our model is superior to MCUNet in both accuracy and memory utilization. The accuracy is boosted to 93% with slightly fewer memory requirements. To be comparable with MCUNetV2, we used the same patch-based inference scheduling to further reduce the memory footprint. Without accuracy loss, the runtime peak memory can be decreased by about $3.5\times$.

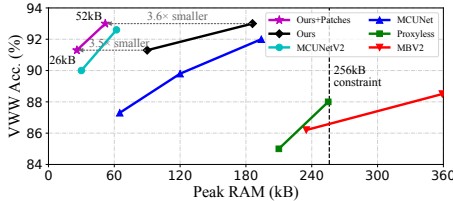

Figure 9: Comparison of different MCU models on Visual Wake Word (VWW) accuracy within 256KB peak memory.

Table 5: Strategy studies for ImageNet classification on IoT devices.

| Model | SRAM Budget, Flash Budget | | |
|---|---|---|---|
| | 256kB, 1MB | 320kB, 1MB | 512kB, 2MB |
| Ours-int8 | 62.7% | 63.9% | 70.4% |
| Ours-int4 | 64.4% | 65.5% | 71.0% |
| Ours-mixed w/o QBR | 65.6% | 66.6% | 71.7% |
| Ours-mixed | **66.5%** | **68.2%** | **72.8%** |

**Resource Maximization.** Table 5 compares our final model with fixed-bit counterparts and shows the effectiveness of QBR. Although 4-bit quantization is better than 8-bit quantization, there is still a gap between fixed- and mixed- precision quantization. Note that even without QBR, our mixed-bit network is superior to fixed-bit MCUNet and MCUNetV2 in Table 4. Fig. 10 visualizes peak memory, bits distribution, and model size of our searched model under 320kB SRAM and 1MB Flash. Comparison of models with or without the QBR is displayed in Figure 11. Obviously, our method maintains higher-precision weights and lower-precision activations in the front few layers, while opposite in the latter few layers. This demonstrates that QBR can strengthen resource utilization by explicitly embedding prior design knowledge.

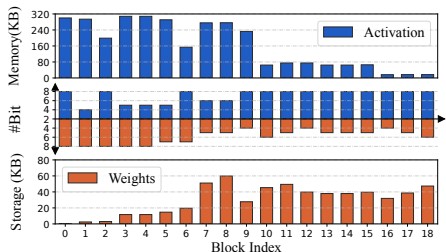 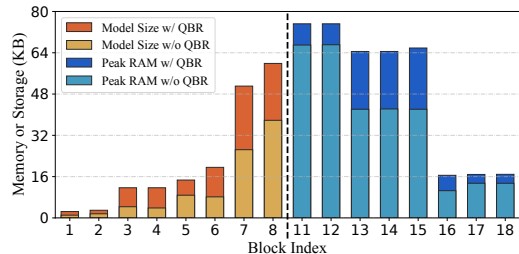

Figure 10: Peak memory, bits distribution and models size of mixed-precision model.

Figure 11: Model size and Peak Memory of models with or without the QBR.

## 5.4 Tiny Object Detection on WIDER FACE

Table 6: Comparison of face detection on WIDER FACE. The hard subset is the most authoritative benchmark since it contains the faces in easy and medium subsets [19].

| Model | Peak RAM | MACs | mAP | | |
|---|---|---|---|---|---|
| | | | Easy | Medium | Hard |
| EagleEye [41] | 1.17MB | 0.08G | 0.74 | 0.70 | 0.44 |
| RNNPool [27] | 1.17MB | 0.10G | 0.77 | 0.75 | 0.53 |
| MCUNetV2 [16] | 762kB | 0.11G | **0.85** | 0.81 | 0.55 |
| Ours-Face | **650kB** | **0.04G** | 0.82 | **0.81** | **0.77** |

The proposed QE-Score method can be adopted to various visual tasks, and we demonstrate its generalization ability on object detection task. Following the pipeline of [19], We conducted experiments on WIDER FACE [37] and results are shown in Table 6. We follow the quantization strategy in [27] that memory usage is analyzed for detector backbone and reported in FP32 to build our detection model "Ours-Face". According to Table 6, we observe that our model can achieve a competitive mAP performance at all three subsets with $1.8\times$ low peak memory and $2\times$ less computation than EagleEye [41] and RNNPool [27], which verifies the generalization ability of our method.

## 6 Conclusion

In this work, we propose an entropy-driven mixed-precision quantization strategy to address challenges of deep network design on IoT devices. In particular, we first present the ranking strategy of mixed-precision quantization networks in the entropy view by formulating the neural network as an information system, and propose a QE-Score with calibrated initialization to measure the expressiveness of the system. Then we propose the Quantization Bits Refinement within evolution algorithm to adjust mixed-precision quantization. Finally, the obtained network achieve state-of-the-art performance on IoT devices for both classification and detection tasks within less than half a 64-core CPU hour searching.

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
