# Entropy-Driven Mixed-Precision Quantization for Deep Network Design: Appendix

## A  Proof of Quantization Entropy Score

### A.1  Maximum Entropy

According to the Maximum Entropy Principle [6] and its successful deep learning applications [17, 1, 25, 19], a deep neural network can be regarded as an information system, and the differential entropy of the last output feature map represents the expressiveness of the system. Given by the following theorem:

**Theorem 1.** *For any continuous distribution $\mathbb{P}(x)$ of mean $\mu$ and variance $\sigma^2$, its differential entropy is maximized when $\mathbb{P}(x)$ is a Gaussian distribution $\mathcal{N}(\mu, \sigma^2)$.*

The differential entropy of a distribution is upper bounded by a Gaussian distribution with the same mean and variance. Suppose $x$ is sampled from Gaussian distribution $\mathcal{N}(\mu, \sigma^2)$, the differential entropy [13] $H$ of $x$ is then:

$$H(x) = \int_{-\infty}^{+\infty} -\log(p(x))p(x)\,dx \quad \propto \log(\sigma^2), \tag{1}$$

where $p(x)$ represents the probability density function of $x$. Moreover, the entropy $H$ represents the expressiveness of a deep system, which correlated with the performance of a deep neural network [19]. Specifically, given a convolutional network with $L$ layers of weights $\mathbf{W}^1, ..., \mathbf{W}^L$, which are initialized from a standard Gaussian distribution, the forward inference is given by:

$$\boldsymbol{x}^l = \phi(\mathbf{W}^l * \boldsymbol{x}^{l-1}) \quad \text{for } l = 1, \ldots, L, \tag{2}$$

where $x^l$ denotes the $l^{th}$ layer feature map and $\phi(\cdot)$ represents the activation function. For simplicity, the bias of the convolutional layer is set to zero and the activation function is omitted. As we initialize the input $\boldsymbol{x}^0$ from Gaussian distribution, the upper bound entropy $H(\mathcal{F})$ of the network $\mathcal{F}$ is proportional to $\log(\sigma^2(\boldsymbol{x}^L))$, where $\sigma^2(\boldsymbol{x}^L)$ represents the variance of the $\boldsymbol{x}^L$ and is computed by the forward inference.

### A.2  Expectation and Variance Theorems

Considering the *product law of expectation* [12]:

**Theorem 2.** *Given two independent random variables $v_1$, $v_2$, the expectation of their product $v_1 v_2$ is:* $\mathbb{E}(v_1 v_2) = \mathbb{E}(v_1)\mathbb{E}(v_2)$.

We can thus compute the variance of the product of these variables as:

$$\sigma^2(v_1 v_2) = \sigma^2(v_1)\sigma^2(v_2) + \sigma^2(v_2)[\mathbb{E}(v_1)]^2 + \sigma^2(v_1)[\mathbb{E}(v_2)]^2, \tag{3}$$

where a simple proof is included in supplementary materials. Meanwhile, the Bienaymé's identity in probability theory [10] is:

**Theorem 3.** *Given $n$ random variables $\{v_1, v_2, ..., v_i, v_{i+1}, ..., v_n\}$ which are pairwise independent integrable, the sums of their expectations and variances are:*

$$\mathbb{E}(\sum_{i=1}^{n} v_i) = \sum_{i=1}^{n} \mathbb{E}(v_i), \quad \sigma^2(\sum_{i=1}^{n} v_i) = \sum_{i=1}^{n} \sigma^2(v_i). \tag{4}$$

### A.3 Variance of One Convolution

Based on the above Theorems 2 and 3, we can obtain the expectation of the element $i$ of $l^{th}$ layer feature map in our system by:

$$\mathbb{E}(\boldsymbol{x}_i^l) = \mathbb{E}(\sum_{h=1}^{K_h^l}\sum_{w=1}^{K_w^l}\sum_{c=1}^{C^{l-1}} \boldsymbol{x}_{chw}^{l-1}\boldsymbol{W}_{chw}^l)$$
$$= \sum_{h=1}^{K_h^l}\sum_{w=1}^{K_w^l}\sum_{c=1}^{C^{l-1}} [\mathbb{E}(\boldsymbol{x}_{chw}^{l-1}) \times \mathbb{E}(\boldsymbol{W}_{chw}^l)], \quad (5)$$

where $\{K_h^l, K_w^l\}$ represents the kernel size of the $l^{th}$ layer in the CNN, and $C^{l-1}$ denotes its input channels size. Note that $C^{l-1}$ is equal to 1 when the layer is a depth-wise convolution. Besides, $t, h, w$ denote the temporal, height and width positions, respectively. As $\boldsymbol{x}^0$ and all parameters $\boldsymbol{W}$ in the system are initialized from a zero-mean Gaussian distribution, which means that $\mathbb{E}(\boldsymbol{W}_{chw}^l) = 0$, which will lead that $\mathbb{E}(\boldsymbol{x}_i^l) = 0$. Then we can obtain the variance of $l^{th}$ layer:

$$\sigma^2(\boldsymbol{x}_i^l) = \sum_{h=1}^{K_h^l}\sum_{w=1}^{K_w^l}\sum_{c=1}^{C^{l-1}} (\boldsymbol{x}_{chw}^{l-1}\boldsymbol{W}_{chw}^l)$$
$$= \sum_{h=1}^{K_h^l}\sum_{w=1}^{K_w^l}\sum_{c=1}^{C^{l-1}} \Big\{\sigma^2(\boldsymbol{x}_{chw}^{l-1}) \times \sigma^2(\boldsymbol{W}_{chw}^l) \quad (6)$$
$$+ \sigma^2(\boldsymbol{x}_{chw}^{l-1})[\mathbb{E}(\boldsymbol{W}_{chw}^l)]^2 + \sigma^2(\boldsymbol{W}_{chw}^l)[\mathbb{E}(\boldsymbol{x}_{chw}^{l-1})]^2\Big\},$$

Since $\mathbb{E}(\boldsymbol{W}_{chw}^l) = 0$ and $\mathbb{E}(\boldsymbol{x}_i^l) = 0$, the variance of element $i$ of $\boldsymbol{x}^1$ layer feature map in our system can be modified to:

$$\sigma^2(\boldsymbol{x}_i^1) = \sum_{h=1}^{K_h^1}\sum_{w=1}^{K_w^1}\sum_{c=1}^{C^0} \Big[\sigma^2(\boldsymbol{x}_{chw}^0) \times \sigma^2(\boldsymbol{W}_{chw}^1)\Big], \quad (7)$$

where $\{K_h^1, K_w^1\}$ represents the kernel size of the first layer in the CNN, and $C^0$ denotes its input channels size. Further extend to the $l$-layer convolution, the variance is:

$$\sigma^2(\boldsymbol{x}_i^l) = \sum_{h=1}^{K_h^l}\sum_{w=1}^{K_w^l}\sum_{c=1}^{C^{l-1}} \Big[\sigma^2(\boldsymbol{x}_{chw}^{l-1}) \times \sigma^2(\boldsymbol{W}_{chw}^l)\Big]. \quad (8)$$

Notice that elements in the same layer feature map have the same distribution with same expectation and variance, then the Eq. 8 can be revised as:

$$\sigma^2(\boldsymbol{x}_i^l) = K_h^l K_w^l C^{l-1}\Big[\sigma^2(\boldsymbol{x}_i^{l-1}) \times \sigma^2(\boldsymbol{W}^l)\Big]. \quad (9)$$

### A.4 Quantization Entropy Score

Mixed-precision quantization to the network $F$ requires inserting low-precision conversion behind Gaussian initialized input and weights. Learning from the practices of entropy coding [18, 13] with fixed quantization step size of 1, given the variable $R$ to quantize symmetrically, the conversion of $N$-bit quantization representation [14, 4] can be defined as:

$$Q = \text{round}(\text{clamp}(R, -2^{N-1}, 2^{N-1} - 1)), \quad (10)$$

where $\text{clamp}$ returns $R$ with values below $-2^{N-1}$ set to $-2^{N-1}$ and values above $2^{N-1} - 1$ set to $2^{N-1} - 1$. According to the work of [19], the input of each layer is zero-mean distribution when deriving the entropy, so that the upper bound of $Q$ is set as $2^{N-1}$. Then, the variance of the

quantization value $Q$ can be given by:

$$\hat{\sigma}^2(N) = \sum \left[\mathbb{P}(Q) \times (Q - \mu_Q)^2\right], \quad \mathbb{P}(Q) = \begin{cases} \int_{-\infty}^{-2^{N-1}+0.5} P(R), & Q = -2^{N-1}, \\ \int_{Q-0.5}^{Q+0.5} P(R), & -2^{N-1} < Q < 2^{N-1}, \\ \int_{2^{N-1}-0.5}^{\infty} P(R), & Q = 2^{N-1}, \end{cases}$$

(11)

where $Q \in [-2^{N-1}, 2^{N-1}]$, $\hat{\sigma}^2$ denotes the truncated variance of $Q$, $R$ obeys $\mathcal{N}(0, \sigma_0)$ and $\sigma_0$ is the initial variance for the variable.

Then, apply the quantization to Eq. 8, the $\sigma^2(x_j)$ of $j^{th}$ layer become larger during the depth increase. As we set the quantization step as 1 in Eq. 10, the distribution of $R$ will be much smoother, and the probability will close to 0. So that the different bits of quantization can not be distinguished in the next layer. If we use a dynamic quantization step in each layer to avoid this phenomenon, the computation will be more complex. Therefore, we adopt a scaling parameter $\sigma_S^2(x_{chw}^{l-1})$ to normalize the $x_j$ to $\sigma_A$ instead of adjusting the quantization step:

$$\sigma_S^2(x_{chw}^{l-1}) = \sigma^2(x_{chw}^{l-1})/\sigma_A^2.$$

(12)

Following the weights obey $\mathcal{N}(0, \sigma_W)$ [9, 19], then use $N_x^{l-1}$ and $N_W^l$ bit to quantize the $l$-layer's input and weight respectively, Eq. 8 can be revised as

$$\begin{aligned} \sigma^2(x_i^l) &= \sum_{h=1}^{K_h^l} \sum_{w=1}^{K_w^l} \sum_{c=1}^{C^{l-1}} \left[\hat{\sigma}^2(N_x^{l-1}) \times \hat{\sigma}^2(N_W^l)\right] \times \sigma_S^2(x_{chw}^{l-1}) \\ &= K_h^l K_w^l C^{l-1} \left[\hat{\sigma}^2(N_x^{l-1}) \times \hat{\sigma}^2(N_W^l)/\sigma_A^2 \times \sigma^2(x_{chw}^{l-1})\right]. \end{aligned}$$

(13)

Subsequently, the variance of $x^L$ can be obtained by propagating the variances from previous layers:

$$\sigma^2(x_i^L) = \sigma^2(x_{chw}^0) \times \prod_{l=1}^{L} K_h^l K_w^l C^{l-1} \hat{\sigma}^2(N_x^{l-1}) \hat{\sigma}^2(N_W^l)/\sigma_A^2,$$

(14)

where $x^0$ is initialized by $\sigma_A^2$. Refer to the Eq. 1, the upper bound entropy $H(\mathcal{F})$ is proportional to:

$$H(\mathcal{F}) \propto \sum_{l=1}^{L} \log\left[K_h^l K_w^l C^{l-1} \hat{\sigma}^2(N_x^{l-1}) \hat{\sigma}^2(N_W^l)/\sigma_A^2\right] + \log(\sigma_A^2),$$

(15)

where $\hat{\sigma}^2(N_x^l)$ and $\hat{\sigma}^2(N_W^l)$ are calculated by Eq. 11 with initial standard deviation $\sigma_A$ and $\sigma_W$. We name the value of Eq. 15 as **Quantization Entropy Score** (QE-Score).

## B  QBR Algorithm

Evolutionary Algorithm (EA) is applied to search neural networks [19] with quantization incorporated into search space. After randomly mutating candidate network, we apply QBR Algorithm 1 to assess the bit distribution and make appropriate adjustments to maximize resources utilization. Firstly, QBR profiles the peak memory of each network block. For each block, the quantization bit is scaled down so that the peak memory can fit in budget, while maximizing quantization precision as much as possible. Since the Flash budget constrains the total weights of all network layers. There are two cases. (i) If the current model size exceeds Flash budget, QBR reduces the weight size from the largest block by randomly decreasing quantization bits until the total model size falls below budget. (ii) If the model size is smaller than the Flash budget, QBR increases quantization precision from the smallest blocks until the total model size approaches the budget. This strategy can improve the utilization of limited resources without affecting the existing evolutionary algorithm, potentially helping to find more expressive structures.

---

**Algorithm 1** Quantization Bits Refinement (QBR)

---

**Require:** Mutated structure $\mathcal{M}$, Quantization space $\mathcal{B}$, Peak RAM budget $\mathcal{R}$, Flash budget $\mathcal{F}$
**Ensure:** Refined structure $\mathcal{M}^{\star}$

1: Get per-block peak memory $\mathbb{Z}$ of $\mathcal{M}$                          ▷ Adjust activation bits
2: **for** $l = 1, 2, \ldots, L$ **do**
3:      Get quantization bit $b_l$ of $\mathbb{Z}_l$
4:      $b_l' = \text{floor\_round}(b_l \times \mathbb{Z}_l / \mathcal{R})$
5:      $b_l = \max(\{\, b \mid b \in \mathcal{B} \text{ and } b < b_l' \,\})$
6: **end for**
7: Get per-block weight size $\mathbb{W}$ of $\mathcal{M}$                    ▷ Redistribute weight bits
8: Sort $\mathbb{W}$ in ascending order
9: $b_{\min}, b_{\max} = \min(\mathcal{B}), \max(\mathcal{B})$
10: **if** $\text{sum}(\mathbb{W}) > \mathcal{F}$ **then**
11:      **for** $l = L, L-1, \ldots, 1$ **do**
12:          Get quantization bit $b_l$ of $\mathbb{W}_l$
13:          $b_l = \text{random\_choice}(\{\, b \mid b \in \mathcal{B} \text{ and } b_{\min} \le b < b_l \,\})$
14:          Recalculate per-block weight size $\mathbb{W}$
15:          **if** $\text{sum}(\mathbb{W}) \le \mathcal{F}$ **then**
16:             **break**
17:          **end if**
18:      **end for**
19: **else if** $\text{sum}(\mathbb{W}) < \mathcal{F}$ **then**
20:      **for** $l = 1, 2, \ldots, L$ **do**
21:          Get quantization bit $b_l$ of $\mathbb{W}_l$
22:          $b_l' = \text{random\_choice}(\{\, b \mid b \in \mathcal{B} \text{ and } b_l < b \le b_{\max} \,\})$
23:          Recalculate per-block weight size $\mathbb{W}'$
24:          **if** $\text{sum}(\mathbb{W}') > \mathcal{F}$ and $\text{sum}(\mathbb{W}) \le \mathcal{F}$ **then**
25:             **break**
26:          **else**
27:             $b_l = b_l'$
28:          **end if**
29:      **end for**
30: **end if**
31: Return $\mathcal{M}^*$ with quantization bits refined to maximize resources

---

## C   Implementation Details

**Searching Settings**. The evolutionary algorithm and super-block definition follow the work of [9, 19]. The evolutionary population $N$ is set as $512$ with total $500000$ iterations. To compare with common mix-quantization search [20, 22, 5, 21] and IoT device applications [8, 7, 15, 24], we build our network with MobileNetV2-based blocks, using the inverted bottleneck block with expansion ratio in $[1.5, 6]$. When starting the search, the initial structure is composed of 5 down-sampling stages with small and narrow blocks to meet the reasoning budget. The width of the selected block is mutated in a given scale $\{2.0, 1.5, 1.25, 0.8, 0.6, 0.5\}$, while the depth of the block increases or decreases 1 or 2. Specifically, low-precision values are random selected from $\{2, 3, 4, 5, 6, 8\}$ in our search, and three layers in each block share the same precision value.

**Datasets and Training Settings**. We use two standard benchmarks in this work: ImageNet [3] and Visual Wake Words (VWW) [2]. For ImageNet-1K dataset, our models are trained for 240 epochs without special indication. All models are optimized by SGD with a batch size of 512 and Nesterov momentum factor of 0.9. Initial learning rate is set to 0.4 with cosine learning rate scheduling [11], and the weight decay is set to 4e-6. Besides, to validate the expansibility of our method, we further evaluate it on the WIDER FACE [23] object detection dataset. When training with low-precision values, we follow the implementation in [4] to do the quantization. We train quantization networks from scratch rather than fine-tuning from full-precision models, saving training time. When training networks with 2-bit activation quantization, the accuracy collapses to NAN easily, whether scratch or fine-tuning. Therefore, we avoid 2-bit quantization for all activations.

## D Limitations

Our approach achieves promising results on IoT devices in the tiny deep model regime, but it still has the following limitations:

- **Finite calibration.** Since the joint search space is huge, we think speed is more important than fitting accuracy at this moment. If we have a deeper study about the distribution of pre-trained weights or tune different Gaussian initialization values for each stage, the correlation between score and accuracy may be better, but this will consume more computational resource and time.

- **Finite search space.** We only evaluate QE-Score in MobileNetV2 search space. Since the motivation is to design a mix-precision network for IoT devices, we choose MobileNetV2 block, the mostly widely used block in mobile or IOT devices, as the search space. Just like the exploration of MAE-DET [19] on ResNet block, we believe the QE-Score can also do that, but it is not the focus of our approach.

- **Search without patch-based strategy.** Patch-based strategy [7] is good for further reducing memory usage. Since the strategy is an orthogonal technique that can be directly applied to the searched models, we didn't integrate it into our search to reduce complexity.

- **Hardware support**. Since not all IoT devices support arbitrary low-precision computation, this means our approach only has advantages in saving SRAM memory and Flash storage, but not in the computation. We hope that our work can demonstrate the power of mixed-precision models and attract the attention of hardware companies to support the low-precision computation.

## E Potential Societal Impacts

**Positive Impacts.** Our approach can achieve architecture searching within less than half a CPU hour, which will bring AI applications to every aspect of our life that rely on IoT devices, including health care, smart agriculture, digital home, smart retail, intelligent transportation, etc.

- **Accessibility.** QE-Score greatly reduces the dependence on hardware and gives more people the opportunity to use NAS algorithms. Additionally, with low cost and many IoT devices, people from rural and under-developed areas can also enjoy the benefits of AI.

- **Privacy protection.** Tiny AI models can be easily deployed on IoT devices, and IoT devices have no need to transmit data to the cloud. Local data processing will reduce the risk of data leakage and helps protect personal privacy.

**Negative Impacts.** The proposed NAS approach is inherently harmless, just like many other basic AI technologies. However, if evildoers apply this technology to malicious applications, such as surveillance, personal information collections, it will generate some negative societal impacts.

- **Accessibility.** The lower cost also gives evildoers more opportunities to create malicious AI programs and deploy them on more portable hardware devices. We advocate using all kinds of AI technologies in a kind way, and not doing evil.

- **Privacy Disclosure.** Accessibility allows more portable hardware to use various AI algorithms, which will also aggravate privacy disclosure. For this, we hope everyone will pay attention to personal privacy and use unknown hardware carefully.

## F Detailed Structure

The searched network structures are listed in Tables 1, 2, and 3. The 'block' column indicates the block type: 'Conv' represents the standard convolutional layer followed by BatchNorm and ReLU, while 'MBConv' is the inverted residual block proposed in MobileNetV2 [16]. 'kernel' and 'stride' are the kernel size and stride size of the depthwise convolution in MBConv. 'in', 'out', and 'btn' represent the channels of input, output, and bottleneck, respectively. '#blocks' is the number of repetitions of the current block. When a block is repeated more than once, starting from the second duplicate, 'stride' will be fixed to 1 and 'in' will be set to 'out' of the previous one. 'nbitsA' and 'nbitsW' represent the quantization bits of activation and weights, respectively.

Table 1: Network structure under 256kB SRAM and 1MB Flash

| block | kernel | stride | in | out | btn | #blocks | nbitsA | nbitsW |
|-------|--------|--------|------|-----|-----|---------|--------|--------|
| Conv | 3 | 2 | 3 | 8 | - | 1 | 8 | 8 |
| MBConv | 5 | 2 | 8 | 32 | 32 | 2 | [4, 8] | [8, 8] |
| MBConv | 5 | 1 | 32 | 48 | 80 | 2 | [5, 8] | [8, 8] |
| MBConv | 5 | 2 | 48 | 96 | 112 | 2 | [4, 8] | [7, 7] |
| MBConv | 5 | 1 | 96 | 96 | 344 | 2 | [6, 6] | [5, 5] |
| MBConv | 5 | 2 | 96 | 96 | 280 | 2 | [7, 8] | [5, 5] |
| MBConv | 5 | 1 | 96 | 128 | 320 | 2 | [8, 8] | [4, 4] |
| MBConv | 5 | 1 | 128 | 128 | 256 | 2 | [8, 8] | [4, 5] |
| MBConv | 5 | 2 | 128 | 160 | 240 | 2 | [8, 8] | [5, 4] |
| MBConv | 3 | 1 | 160 | 160 | 192 | 2 | [8, 8] | [5, 5] |

Table 2: Network structure under 320kB SRAM and 1MB Flash

| block | kernel | stride | in | out | btn | #blocks | nbitsA | nbitsW |
|-------|--------|--------|------|-----|-----|---------|--------|--------|
| Conv | 3 | 2 | 3 | 16 | - | 1 | 7 | 8 |
| MBConv | 5 | 2 | 16 | 32 | 32 | 2 | [4, 8] | [8, 8] |
| MBConv | 5 | 1 | 32 | 32 | 128 | 2 | [5, 5] | [8, 8] |
| MBConv | 5 | 2 | 32 | 80 | 120 | 2 | [5, 8] | [7, 7] |
| MBConv | 5 | 1 | 80 | 120 | 360 | 2 | [6, 6] | [5, 5] |
| MBConv | 5 | 2 | 120 | 152 | 184 | 2 | [8, 8] | [4, 6] |
| MBConv | 5 | 1 | 152 | 152 | 240 | 2 | [8, 8] | [5, 4] |
| MBConv | 5 | 1 | 152 | 152 | 184 | 2 | [8, 8] | [5, 5] |
| MBConv | 5 | 2 | 152 | 152 | 192 | 2 | [8, 8] | [5, 4] |
| MBConv | 3 | 1 | 152 | 160 | 192 | 2 | [8, 8] | [5, 6] |

Table 3: Network structure under 512kB SRAM and 2MB Flash

| block | kernel | stride | in | out | btn | #blocks | nbitsA | nbitsW |
|-------|--------|--------|------|-----|-----|---------|--------|--------|
| Conv | 3 | 2 | 3 | 16 | - | 1 | 8 | 8 |
| MBConv | 5 | 2 | 16 | 64 | 64 | 2 | [4, 8] | [8, 8] |
| MBConv | 5 | 1 | 64 | 72 | 192 | 2 | [5, 5] | [8, 8] |
| MBConv | 5 | 2 | 72 | 72 | 192 | 2 | [5, 8] | [8, 8] |
| MBConv | 5 | 1 | 72 | 72 | 960 | 2 | [5, 5] | [5, 5] |
| MBConv | 5 | 2 | 72 | 120 | 720 | 2 | [6, 8] | [5, 5] |
| MBConv | 5 | 1 | 120 | 160 | 576 | 2 | [8, 8] | [5, 5] |
| MBConv | 5 | 1 | 160 | 160 | 600 | 2 | [8, 8] | [5, 4] |
| MBConv | 5 | 2 | 160 | 160 | 536 | 2 | [8, 8] | [5, 5] |
| MBConv | 3 | 1 | 160 | 160 | 760 | 2 | [8, 8] | [4, 5] |