# OpenReview forum: "Entropy-Driven Mixed-Precision Quantization for Deep Network Design"
_NeurIPS.cc/2022/Conference — NeurIPS 2022 Accept_

### Official Review · Reviewer_TPBM · 2022-07-11

**Rating:** 8
**Confidence:** 5
**Soundness:** 3 good
**Presentation:** 4 excellent
**Contribution:** 4 excellent

**Summary:**

The paper proposes a new mixed-precision quantization in a new perspective of information entropy. Since the proposed Quantization Entropy Score can approximately measure the expressiveness of neural networks and is easy to compute, the authors can do the search in a short time.

**Questions:**

1. Since the search cost is very low, why do the authors expand the search space? We may expect to find better results in a larger solution space. For instance, could you relax the constraints mentioned in Line 186 (three layers in each block share the same precision value)?
2. In Table 5, why does the QBR achieve better results? From my understanding, the QBR uses prior knowledge to shrink the search space. Without QBR, we should explore a larger space and achieve better results than that with QBR.
3. Could the authors discuss other hardware-related metrics, such as latency, throughput, and energy?


**Limitations:**

Limitations and social impacts are well discussed in Appendix E and F.

**Strengths And Weaknesses:**

Strengths
1. The paper is well developed and organized. The tables and figures are great.
2. The authors extend the previous work of viewing neural networks from the perspective of information entropy. Since quantization can be treated as an operator, it is natural to apply this perspective to the quantization problem.
3. The experiments are convincing. The authors cover three benchmarks.

Weaknesses
1. I am wondering if there is any work on using the entropy-based scores to conduct neural architecture search (NAS). If yes, the related work should be discussed and compared. If no, it should be a contribution of this paper. The authors should also show that the score can be used to search for a full precision model efficiently.
2. How can the proposed method extend to other machine learning models, such as Transformers?

---

> ### Author Response · Authors · 2022-08-02
> **Response to Reviewer TPBM**
>
> Thanks for all your constructive comments. Please see below our response to the specific questions.
>
> > Q1: Entropy-based related work
>
> A1: As mentioned in Line 39-42 and Section 3.1, our proposed QE-score is inspired by a full precision entropy-based method, MAE-DET [1].
> Compare with MAE-DET, we apply formula calculation rather than inference (Section 3.1). Moreover, we expand this theory to a mixed-precision quantization network (Sections 3.2 and 3.3). Appendix A discussed the difference between our method and related work. In addition, we add an analysis of the difference between entropy score in MAE-DET with QE-score ($\sigma_A=\sigma_W=1$), as shown in Appendix H of the revised version. Please have a look.
>
> > Q2: How can the proposed method extend to other machine learning models, such as Transformers?
>
> A2: We think the Maximum Entropy Principle is applicable theoretically to Transformers. However, there exist some challenges to overcome. For example, Transformer has more complex components than CNN, such as 'Q, K' kernel operation and multi-head attention, which is difficult to calculate the maximum entropy.  Although these challenges are difficult to overcome, it would be an interesting task for us in future work.
>
> > Q3: Large search space
>
> A3: Thank you for your suggestion. We follow the settings of the Ours-19.2G model to conduct an experiment by relaxing the precision value on three layers. TOP-1 accuracy is 75.1\% with 18.8G BitOps and 3.4M Params, which is slightly better than the Ours-19.2G model. According to the result, we can see that large search spaces actually benefit the performance. However, compared with the results in Table 3 (MobileNetV2 and ours mixed), the mixed-precision quantization strategy can boost the performance (+3\%) more than a large search space did (+0.2\%). Due to limited rebuttal time, we will explore the detailed large search space discussion in the next version.
>
> > Q4: Why does the QBR achieve better results?
>
> A4: QBR shrinks the search space, but it improves resource utilization based on the consensus of higher bit is better in constrained environments. To make full use of the resource budget, we propose the QBR to achieve a robust search result for low-resource IoT devices.
>
> > Q5: Could the authors discuss other hardware-related metrics, such as latency, throughput, and energy?
>
> A5: For different target hardware, our current model (i.e. Ours-19.2G) may not achieve optimal performance under these metrics. Our method focuses on mixed-precision quantization network design, and our method can achieve an optimal architecture under these metrics. However, deploying searched models to real hardware requires more time. Due to limited time, we will add a detailed discussion on hardware-related metrics in the next version.
>
> [1]: Zhenhong Sun, Ming Lin, Xiuyu Sun, Zhiyu Tan, Hao Li, and Rong Jin. MAE-DET: Revisiting Maximum Entropy Principle in Zero-Shot NAS for Efficient Object Detection. *In International Conference on Machine Learning*, pp. 20810-20826, 2022.

---

### Official Review · Reviewer_Aoyu · 2022-07-11

**Rating:** 6
**Confidence:** 3
**Soundness:** 3 good
**Presentation:** 3 good
**Contribution:** 3 good

**Summary:**

Previous works use an inefficient two-stage method that quantizes after designing a small network. However, the proposed method uses a one-stage method that can jointly optimize network design and mixed-precision quantization. Therefore, it is not only efficient in terms of time-cost, but also achieved high performance in various tasks.

**Questions:**

1) Can you provide applicability and experimental results for networks other than MobileNet-V2?
2) Can you show additional experimental results related to Weaknesses 2), 3) and 5)?
3) Can you provide additional experimental results and explanations regarding the compatibility of the proposed method?
4) Can you provide additional explanations related to Weaknesses 6) and 7)?

**Limitations:**

The authors have adequately addressed the limitations and potential negative societal impact of their work.

**Strengths And Weaknesses:**

Strengths:
1) The ranking strategy from the perspective of entropy and Quantization Bits Refinement in the evolutionary algorithm considering the HW budget are interesting and novel.
2) This paper clearly defined the problems and demonstrated solutions through solid mathematical analysis.
3) This paper is very clear and easy to read. I think that it would help both researchers related to this field and general readers to give a little bit more background on TinyML.
4) The experiment showed the superiority of the proposed method on various datasets (ImageNet, VWW, WIDER FACE).

Weaknesses:
1) For all experiments, MobileNet-V2 was used as the base model. In particular, the analysis of peak memory and model size presented in Section 2 is also limited to MobileNet-V2. However, when a different model is used, the peak memory for each layer will also be different. Can we assume that the results of MobileNet-V2 are representative? The authors should show that the proposed method is compatible with other models.
2) Since the key contribution asserted in this paper is the 1-stage approach, it is necessary to thoroughly compare the proposed method with the 2-stage methods. In other words, comparative experiments with more combinations of SOTA network design studies + SOTA mixed-precision studies should be presented.
3) Experimental results in terms of power (energy) consumption should be presented. As the authors know, the most important factor in an IoT device is energy consumption.
4) It is necessary to explain whether the proposed method is compatible with more complex networks (e.g., SSD, instance segmentation, Transformer). (If possible, the experimental results need to be presented together.) Also, it is difficult to understand which network was used for the current Tiny Object Detection experiment.
5) [1] was used as a quantization-aware training (QAT) method. However, model performance may be improved by [1] in this paper. So, the current results cannot clearly show the effectiveness of the proposed method. Further experiments excluding the impact of QAT should be presented.
6) In this paper, only SRAM and Flash are used as constraints, but it seems that DRAM should be considered.
7) Why didn't the authors use the commonly used bit-precision configuration {2, 4, 8, 16} in the mixed-precision setting? Additional explanations related to this should be provided.
8) In Table 3, the appropriate reference should be added on which tool was used to measure CO2 emission.

[1] Learned step size quantization. ICLR’20.

---

> ### Author Response · Authors · 2022-08-02
> **Response to Reviewer Aoyu (1/2)**
>
> Thanks for all your constructive comments. Please see below our response to the specific questions.
>
> > Q1: Can we assume that the results of MobileNet-V2 are representative? The authors should show that the proposed method is compatible with other models.
>
> A1: We conduct a brief experiment on the ResNet-50 model. The results are displayed below, which verify that our QE-Score still works well on the ResNet series model. We will explore the detailed experiments in the next version.
>
> | Model | Quant. | Model Size (MB) | BitOps (G) | ImageNet TOP-1 |
> | :--- | --- | --- | --- | --- |
> | ResNet-50 | 8-bit | 23.5 | 262.4 | 78.0\% |
> | ResNet-50 | 4-bit | 13.8 | 72.6 | 76.3\% |
> | Ours | mixed | 25.8 | 261.6 | **79.2\%**|
> | Ours | mixed | 10.9 | 71.0 | **77.6\%**|
>
> Trianing settings: SGD optimizer with momentum 0.9; weight decay 4e-6 for ImageNet; a batch size of 512; initial learning rate 0.1 with 120 epochs.
>
> > Q2: Compare the proposed method with other 2-stage methods
>
> A2: Since there exist few combinations of 2-stage methods, we've updated the results table as shown below. In addition, “Ours-Stage1" model with 8-bit precision value is the first stage searched model, and the "Stage2" model is the second stage model directly quantized with our mixed-precision method. Results show that our 2-stage QE-Score obtains competitive results with SOTA 2-stage methods. Due to the limited rebuttal time, we will add more 2-stage results in the next version. According to the table, we can see that our method still achieves competitive results.
>
> | Model | Quant. | Search Devices | Design Cost (hours) | Model Size (MB) | BitOps (G) | TOP-1 (ImageNet)| CO2e (marginal) |
> | :--- | --- | --- | --- | --- | ---| ---| ---|
> | MBV2 | 8-bit | - | - | 3.4 | 19.2 | 71.9\% | - |
> | MBV2 | 4-bit | - | - | 2.3 | 7.0 | 68.9\% | - |
> | ProxylessNAS[1] | 8-bit | GPUs | 200N | - | 19.5 | 74.2\% | 56.72N |
> | ProxylessNAS + AMC | 8-bit | GPUs | 204N | - | 15.0 | 73.3\% | 57.85N |
> | ProxylessNAS + AMC + HAQ | mixed | GPUs | 300N | - | - |71.8\% | 85.08N|
> | MBV2+HAQ | mixed | GPUs | 96N | - | - | 71.9\% | 27.23N |
> | DNAS | mixed  | GPUs | 300N | - | 57.3 | 74.0\% | 11.34N |
> | SPOS | mixed  | GPUs | 288+24N | - | 51.9 | 74.6\% | 82+6.81N |
> | APQ | mixed   | GPUs | 2400+0.5N | - | 16.5 | 74.1\% | 672+0.14N|
> | Ours-Stage1 | 8-bit | CPUs | 0.5N |2.6 | 19.2 | 73.7\% | 0.19N|
> | Ours-Stage2 | mixed | CPUs | 0.5N | 2.0 | 6.8 | 69.8\% | 0.19N|
> | Ours-19.2G | mixed | CPUs | 0.5N | 3.2 | 18.8 | **74.8\%** | 0.19N |
> | Ours-7.0G | mixed | CPUs | 0.5N | 2.2 | 6.9 | **70.8\%** | 0.19N|
>
> > Q3: Experimental results in terms of power (energy) consumption should be presented.
>
> Our method focuses on mixed-quantization architecture design, and deploy searched models to real hardware requires more time. Due to limited time, we will add a detailed discussion of power consumption in the next version.
>
> > Q4: Whether the proposed method is compatible with more complex networks?
>
> A4: We follow the quantization strategy in [2] that memory usage is analyzed for detector backbone and reported in FP32 to build our detection model 'Ours-Face'. 'Ours-Face' model is proposed to verify the generalization ability of our method, and we will release the code and model. According to the tiny object detection experiment, we think our method is compatible with more complex networks. However, lots of work needs to be done, and our work focuses on investigating a mixed-precision quantization approach under the maximum entropy principle. Due to limited rebuttal time, we will explore more complex networks in future work.
>
> > Q5: The current results cannot clearly show the effectiveness of the proposed method.
>
> A5: The MBV2 models with 8-bit or 4-bit precision values in Table 3 are also implemented with LSQ training methods like our methods. The accuracy of MBV2-8bit is 71.9\%, consistent with the value of 71.8\% in APQ[3], which can help us exclude the model performance improved by the quantization-aware training strategy.

---

> > ### Author Response · Authors · 2022-08-02
> > **Response to Reviewer Aoyu (2/2)**
> >
> > > Q6: In this paper, only SRAM and Flash are used as constraints, but it seems that DRAM should be considered.
> >
> > A6: Our NAS search is based on the peak memory constraints, which are not directly with specific hardware, whether the hardware is SRAM or DRAM. SRAM is specially emphasized for its common use on MCU and lower power than DRAM, which is also aligned with MCUNetV1/V2 for a fair comparison.
> >
> > > Q7: Why didn't the authors use the commonly used bit-precision configuration {2, 4, 8, 16} in the mixed-precision setting?
> >
> > A7: Our work focuses on theoretically exploring mixed-precision quantization network design. There are two main aspects for consideration of bit-precision configuration. Firstly, the accuracy of models with 8-bit precision values is consistent with 16 or 32-bit precision values, which means 8-bit precision values could satisfy the deployment requirements. Secondly, most low-power IoT devices consist of cheap and low-computation-power chips, whose strongest computation precision is 8-bit.
> >
> > > Q8: In Table 3, the appropriate reference should be added on which tool was used to measure CO2 emission.
> >
> > A8: The calculation of CO2 emission follows the APQ [3], in which the CO2 emission is positively correlated with power consumption. The power of GPU used in APQ is about 300W, and that of CPU in our QE-Score is about 400W.
> >
> > [1]: Han Cai , Ligeng Zhu, and Song Han. ProxylessNAS: Direct Neural Architecture Search on Target Task and Hardware. *International Conference on Learning Representations*. 2018.
> >
> > [2]: Saha, O., Kusupati, A., Simhadri, H. V., Varma, M., & Jain, P. RNNPool: efficient non-linear pooling for RAM constrained inference. Advances in Neural Information Processing Systems, 33, 20473-20484, 2020.
> >
> > [3]: Tianzhe Wang, Kuan Wang, Han Cai, Ji Lin, Zhijian Liu, Hanrui Wang, Yujun Lin, and Song Han. APQ: Joint search for network architecture, pruning and quantization policy. *In Proceedings of the IEEE/CVF Conference on Computer Vision and Pattern Recognition*. 2020: 2078-2087.

---

### Official Review · Reviewer_xZvK · 2022-07-11

**Rating:** 5
**Confidence:** 3
**Soundness:** 3 good
**Presentation:** 2 fair
**Contribution:** 3 good

**Summary:**

This paper presents an entropy-driven mixed-precision quantization strategy to design efficient CNNs for resource constrained IoT devices. The paper proposes a new metric, Quantization Entropy Score (QE-Score), to evaluate the CNN model capacity without the need of training the model. QE-Score is applicable when different layers are quantized to different bit widths, and thus can significantly speedup the design space exploration for efficient models. Extensive experiments have been performed to show that the proposed quantization strategy discovers tiny models with higher accuracy under strict resource constraints.

**Questions:**

Please clarify the questions regarding the technical descriptions (see comments on “Quality”).

Please provide more details on the main algorithm steps (e.g., Quantization bits refinement, Evolution algorithm).


**Limitations:**

The paper does not discuss potential negative societal impact.

**Strengths And Weaknesses:**

### Originality

While the connections between deep learning models and information systems have been studied and the entropy has been related with accuracy in the literature, it appears to be an original idea to evaluate CNN model capacity via a single QE-Score metric under the quantization scenario.

The discussion on the relation among quantization budget, neural architecture, initialization and entropy (accuracy) is original and useful.


### Quality

**Strengths**

The proposed QE-Score seems to be a simple yet effective metric to estimate the model capacity. This shows potential for practical IoT deployment.

The evaluation of the proposed strategy is extensive and covers various aspects.

Necessity of mixed-precision quantization is clearly illustrated by the MobileNet layer breakdown in Fig 2.


**Weaknesses**

Various technical descriptions in Section 3 are unclear, and I am challenged to interpret some of the plots. For example:
* Equation 6: what variable is the summation applied on? Q based on its different value ranges?
* Fig 6: it appears to me most subplots can show positive correlation between QE-Score and accuracy. What is so special about the plot of $\sigma_A=5$ and $\sigma_W=4$ plot? Why do you say that "QE-Scores are gradually positive(ly) correlated with accuracy until $\sigma_A=\sigma_W=4$ Then, we adjust $\sigma_A=5$ and $\sigma_W=4$ to rank the diversity of activations and weights on accuracy"? What do you mean by ranking the diversity of activations and weights?
* The Quantization Bits Reinforcement (QBR, Fig 7) strategy is very unclear. Where do you explain this figure? Why do you need this refinement when you already have QE-Score to guide the allocation of bit-widths? You should at least briefly introduce this algorithm in the main text.

The fundamental assumption on the system is unclear: To execute the CNN on IoT devices, do you need the full model to fit in its memory? Or do you only need a single layer (the currently executed layer) to fit in the memory? From Fig 2, it seems that you are imposing a layer-wise constraint. From other parts of the paper, it seems that the constraints are on the full CNN.

From the description, it seems that the main purpose of QE-Score is to simplify neural architecture search (e.g., channel widths, bit widths). Then why does initialization of X and W matter? Specifically, since you will train the model after fixing the architecture, the effect of different initialization scales will be automatically addressed by batch-norm or layer-norm.

While the empirical results seem to justify QE-Score, its derivation seems problematic. Activation has a major impact on the distribution. For example, with ReLU, half of values will be cropped to 0 if the input follows 0-mean Gaussian. With a deep CNN, the final layer distribution will be far from Gaussian. Thus the derivation of QE-Score does not follow a realistic assumption.


### Clarity

The initial portion of the paper is clear to motivate mixed-precision quantization. However, the technical description is overall unclear (see details above).

Tables and figures in experiments clearly show the advantages of the proposed strategy over the baseline w.r.t. various different metrics.


### Significance

Judging from the empirical gains, the proposed method makes a significant contribution to enable efficient IoT inference with compressed models.

However, evaluations are mostly performed on a single backbone model (MobileNet). It is unclear if the proposed method can generalize to other CNN or non-CNN models.

The technical significance seems to be limited. First of all, the QE-Score and the proposed algorithm are both mostly heuristic based. Secondly, due to the questions on the technical sections (listed above), it is unclear if the proposed method brings in-depth insights to the community.

---

> ### Author Response · Authors · 2022-08-02
> **Response to Reviewer xZvK (1/2)**
>
> Thanks for all your constructive comments. Please see below our response to the specific questions.
>
> > Q1: Equation 6: what variable is the summation applied on? Q based on its different value ranges?
>
> A1: The summation in Equation 6 represents the variance of the quantization value Q, and Q is the quantization variable of R.
> Besides, the detailed computation process of the different value ranges of Q is described in Lines 126-128.
>
> > Q2: What is so special about the plot?
>
> A2: As mentioned in Lines 156-163, the two insights are derived from Table 1, Table 2 and Figure 5: 1) Increasing the initial standard deviation $\sigma$ will help us distinguish the disparity of entropy between different bits; 2) Quantization of activations and weights has different effects on accuracy. For example, A3W4 and A4W3 models have the same BitOps, but A4W3 models have about 2\% accuracy higher than A3W4, which means that quantization on activations and weights has different effects on accuracy. A3W6 and A6W3 models have the same situation. Therefore, we set different values of $\sigma_A$ and $\sigma_W$ to show differences between activations and weights.
>
> > Q3: QBR strategy is not clear.
>
> A3: We revised the caption of Figure 7 and a brief description of QBR lines 178-181. The detailed description and algorithm are in Appendix C. In addition, we add an analysis of QBR and w/o QBR models in Appendix C of the revised version. Please have a look.
>
> > Q4: The fundamental assumption of the system
>
> A4: There are two main budget constraints on IoT devices, i.e., memory and storage. Storage is a constraint of the full model and memory is a block-wise constraint, which is widely adopted in previous tinyML literature, such as MCUNet V1 and V2.
>
> > Q5: Why does initialization of X and W matter?
>
> A5: It is worth noting that searching and training models don't use the same initialization. The initialization of X and W matter in searching because we design a QE-Score for measuring the expressiveness of the mixed-precision quantization model. When training, X is not needed to be initialized, and W uses the default Kaming initialization.
>
> > Q6: The derivation of QE-score seems problematic.
>
> A6: The bias of the convolutional layer is set to zero and the activation function is omitted in searching for simplification (Lines 103-108), following the work of ZenNAS [1] and MAE-DET [2] which has been proved that this will not affect the expressiveness of the network. The training of CNN models has been well studied, and some components can be integrated to boost performance. We deliberately avoid using these components to keep our design simple and universal. Nevertheless, these auxiliary components can be easily plugged into the architecture without any special modification.
>
> > Q7: The derivation of QE-Score does not follow a realistic assumption.
>
> A7: As mentioned in Theorem 1 (lines 95-96), for any continuous distribution with specific mean and variance, its differential entropy is maximized when the distribution is a Gaussian distribution, and it only depends on the variance (lines 96-98): $H(F) \propto log(\sigma^2)$; According to the Maximum Entropy Principle (Lines 90-92), the expressiveness of a continuous random distribution is positively correlated with its differential entropy. Similarly, if we regard a convolution neural network as such a system, the expressiveness of the output feature map is maximized when its entropy achieves maximum;
> Furthermore, the distribution of the output feature map is high-dimensional. We assume it obeys the Gaussian distribution to estimate the upper bound of the entropy, which represents its expressiveness (Lines 103-108).

---

> > ### Author Response · Authors · 2022-08-02
> > **Response to Reviewer xZvK (2/2)**
> >
> > > Q8: It is unclear if the proposed method can generalize to other CNN or non-CNN models.
> >
> > A8: We've conducted an experiment about the ResNet-50 model. The results are shown in **A1 of Response to Reviewer Aoyu**.
> > We think the Maximum Entropy Principle is applicable to Transformers theoretically. However, there exist some challenges to overcome. For example, Transformer has more complex components than CNN, such as 'Q, K' kernel operation and multi-head attention, which is difficult to calculate the maximum entropy. Although these challenges are difficult to overcome, it would be an interesting task for us in future work.
> >
> > > Q9: Please provide more details on the main algorithm steps.
> >
> > A9: The detailed QBR algorithm is displayed in Algorithm 1 of Appendix C.
> >
> > > Q10: The paper does not discuss potential negative societal impact.
> >
> > A10: As mentioned in the checklist, limitations and social impacts are discussed in Appendix E and F.
> >
> > [1]: Ming Lin, Pichao Wang, Zhenhong Sun, Hesen Chen, Xiuyu Sun, Qi Qian, Hao Li, and Rong Jin. Zen-nas: A zero-shot nas for high-performance image recognition. *In Proceedings of the IEEE/CVF International Conference on Computer Vision* pp. 347-356, 2022.
> >
> > [2]: Zhenhong Sun, Ming Lin, Xiuyu Sun, Zhiyu Tan, Hao Li, and Rong Jin. MAE-DET: Revisiting Maximum Entropy Principle in Zero-Shot NAS for Efficient Object Detection. *In International Conference on Machine Learning*, pp. 20810-20826, 2022.

---

> ### Comment · Reviewer_xZvK · 2022-08-09
> **Update from Reviewer**
>
> I sincerely thank the authors for providing detailed response. Most of my concerns have been addressed and thus I decide to increase my score.

---

> > ### Author Response · Authors · 2022-08-09
> > **Response to Reviewer xZvK**
> >
> > Dear Reviewer xZvK,
> >
> > Thanks again for all your constructive comments and response.
> >
> > Best regards,
> >
> > Authors

---

### Official Review · Reviewer_gBWZ · 2022-07-12

**Rating:** 7
**Confidence:** 4
**Soundness:** 3 good
**Presentation:** 3 good
**Contribution:** 3 good

**Summary:**

This paper proposes an entropy-driven mixed-precision quantization strategy. The basic idea is to formulate the neural network as an information system, and then define a quantization entropy score to capture the expressiveness of the system. Finally, the Quantization Bits Refinement algorithm quantizes the activation and weights to meet the peak memory and model size constraint.

**Questions:**

1. The QBR strategy looks heuristic, is it possible to fall into an oscillation process?
2. Why only 8 bit and 2 bit in Figure 7?

**Limitations:**

The proposed QE-scores can only use CPU resources instead of GPU.

**Strengths And Weaknesses:**

Strength:
1. The concept of the quantization entropy score looks interesting.
2. The proposed solution outperforms SOTA works.

Weakness:
1. The proposed solution does not work for GPU.

---

> ### Author Response · Authors · 2022-08-02
> **Response to Reviewer gBWZ**
>
> Thanks for all your constructive comments. Please see below our response to the specific questions.
>
> > Q1: The proposed solution does not work for GPU.
>
> A1:  The description might cause you misunderstanding, our proposed solution can work for both GPU and CPU. We've revised this description, and please see lines 212-214 in the revised version. Compare with previous NAS methods, the proposed QE-score calculates the expressiveness of a network through formulas rather than inference, to obtain the optimal network architecture. It will reduce computational complexity, and then speed up the searching time. We also add an analysis of the difference between our QE-score and inference methods in Appendix H of the revised version. Therefore, we can simply apply the CPU platform to calculate QE-score instead of the GPU platform.
>
> > Q2: The QBR strategy looks heuristic, is it possible to fall into an oscillation process?
>
> A2: The below table shows the detail of the QBR process, that we can see QBR strategy is not an oscillation process.
> | Iteration | 5W | 10W | 15W | 20W |  25W |  30W | 35W | 40W |  45W|  50W |
> | :----------| ----   |   ----   |  ----   |  ----   |  ----   |  ---- | ---- | ---- |---- |---|
> | Params (M) |  1.8 |   1.7 |  2.7 |  2.0 |  1.9 |  1.9 |  1.9 |  2.1 |  2.0 |  1.9|
> | BitOps (G) |  3.54 |  4.06 |  6.88 |  6.83 |  6.84 |  6.83 |  6.93|  6.93 |  6.97 |  6.93|
> | Score | 1162.71 |  1296.29 | 1324.55 |  1352.67 |  1364.97 |  1372.77 |  1375.94 |  1377.51 |  1380.36 |  1380.98|
>
> > Q3: Why only 8 bit and 2 bit in Figure 7?
>
> A3: We've revised the caption of Figure 7  and please have a look at the revised paper. Under different budgets, orange curve arrows in the four corners of the figure mean the adjustment scale of the precision value from 2 bit to 8 bit and 8 bit to 2 bit, not just 2bit and 8bit.

---

### Author Response · Authors · 2022-08-02
**General Response**

Dear reviewers and meta reviewers,

We would like to thank all reviewers for their constructive comments and efforts to help improve our work. Considering all the comments, we have thoroughly revised the manuscript. The changes have been highlighted in blue font in the revised paper.

Detailed Q&As are listed below. We look forward to further discussions and feedback.

---

### Author Response · Authors · 2022-08-08
**Response in Discussion Period**

Dear reviewers and meta reviewers:

Hope this message finds you well.

We have answered your questions and updated the paper accordingly. Since the discussion period will end in less than two days, we would like to kindly ask whether you have further concerns or questions that we might be able to address.

Thanks so much!

Best regards,

Authors

---

### Meta-Review · Area_Chair_XAac · 2022-08-28

**Recommendation:** Accept
**Confidence:** Certain

**Metareview:**

this paper proposes a joint architecture design and quantization method by casting it as an Entropy Maximization process. reviewers give consensus acceptance.

**Award:**

No

---

### Decision · Program_Chairs · 2022-09-14

Accept